# Predation Rate on Olive Riley Sea Turtle (*Lepidochelys olivacea*) Nests with Solitary Nesting Activity from 2008 to 2021 at Corozalito, Costa Rica

**DOI:** 10.3390/ani13050875

**Published:** 2023-02-28

**Authors:** Nínive Espinoza-Rodríguez, Daniela Rojas-Cañizales, Carmen Mejías-Balsalobre, Isabel Naranjo, Randall Arauz

**Affiliations:** 1Rescue Center for Endangered Marine Species (CREMA), Nandayure 50906, Costa Rica; 2Grupo de Trabajo en Tortugas Marinas del Golfo de Venezuela (GTTM-GV), Maracaibo 4002, Venezuela; 3Red de Investigadores Actuando por el Medio Ambiente (RIAMA), C/Nuñez de Balboa 114, 28006 Madrid, Spain; 4Marine Watch International, San Francisco, CA 94129, USA

**Keywords:** nest predation, sea turtles, nesting beach, eastern pacific

## Abstract

**Simple Summary:**

Understanding the mechanisms of predation dynamics in a sea turtle nesting beach is important in order to design a proper beach management plan and a valuable conservation strategy when working on sea turtle solitary nesting sites. We evaluated natural nest predation upon solitary Olive Ridley nesting events in Corozalito beach, a solitary and nascent arribada nesting beach on the Pacific in Costa Rica. Our results show a noticeable increase in predation rates close to 30% of nests predated throughout a 13-year study period. This could be an indicator of the increasing number of nesting events in this important nesting site. We suggest continuing to monitor the nesting activity at Corozalito, including the predation rates from other sea turtle species and mass nesting events to give a complete report of the threats that these species are facing in Corozalito and the impact on their population, combined with potential alternatives to manage predators’ impact.

**Abstract:**

In Corozalito beach, Costa Rica, Olive Ridley turtles (*Lepidochelys olivacea*) nest both solitarily and in arribadas. The predation of solitary nests was monitored from 2008 to 2021, recording date, time, sector of the beach, zone, status of nest (predated or partially predated) and predator when possible. We recorded 4450 predated nests in total (N = 30,148 nesting events); predation rates showed a fluctuating trend, with recent percentages reaching up to 30%, with four distinctive dips in 2010, 2014, 2016 and 2017. The spatial distribution of predated nests along the beach showed significant differences among the sectors regardless of the seasons (Friedman test, chi-squared = 14.778, df = 2, *p*-value = 0.000), with most predated nests (47.62%) occurring in the northern sectors of the beach. Predators were identified by their tracks and/or direct observations (N = 896, 24.08%). The most conspicuous predators identified were raccoons (55.69%) and black vultures (22.77%). As seen in Corozalito, predation rates have increased in recent years despite established conservation efforts. A comprehensive assessment of all threats towards the overall hatching success for clutches is needed, considering predation during mass nesting events, poaching and beach erosion, among other factors, to fully understand the nesting dynamics occurring in this beach.

## 1. Introduction

The Olive Ridley sea turtle (*Lepidochelys olivacea* Eschscholtz 1829) is widely distributed in circumtropical waters and reaches its greatest abundance along the eastern rim of the Pacific Ocean [1,2]. It is known to be the most globally abundant sea turtle, being one of the two species in the genus *Lepidochelys* that nests in a unique massive synchronous fashion known as “arribada”, with more than half a million female nesting turtles during a single season [3,4,5,6,7]. Arribadas have been described in the Eastern Pacific, the Western Atlantic, and the Northern Indian Ocean [8,9,10,11,12], but the species also nests widely and abundantly throughout the beaches of the Central American coast in solitary events [1,2,6]. Despite these large numbers and wide global distribution, the Olive Ridley sea turtle is currently listed as Vulnerable by the International Union for the Conservation of Nature’s Red List of Threatened Species with a persistent decreasing population trend [2]. In addition, the species is included in Appendix I of CITES, since it is considered an endangered species whose marketing control (in its entirety or for by-products) is quite strict [13]. Simultaneously, Costa Rica protects Olive Ridley turtle populations under the Wildlife Conservation Law Nº 7317 [14], Protection and Conservation of Sea Turtles Nº 8325 [15] and lists of national species (resolution Nº 092-2017) [16]. The intense protection, either at a national or international level is due to the major cause of detriment to Olive Ridley turtle populations in the Eastern Pacific, which is thought to have been massive commercial overexploitation of their eggs, and to a lesser degree high coastal fishery induced mortality [2,17,18].

The protection of sea turtle nesting beaches to enhance recruitment is a common strategy to assist in the recovery of their populations [19]. Natural stressors, however, play an important role when resorting to this strategy, as sea turtle nests are subject to a series of natural threats such as habitat destruction caused by erosion resulting from high tides, sea level rise, or flash floods at river mouths [5,19]. Nests and hatchlings are also highly vulnerable to predation, most frequently by ants, beetles, snakes, coatis, raccoons, pigs, foxes, coyotes, and domestic animals (e.g., dogs) [20,21]. Secondary or opportunistic predation also occurs in previously raided nests by vultures, crabs, and maggots [22].

The contribution of each predator to the estimated predation rate differs greatly between sites. Historic predation by raccoons (*Procyon lotor*, Linnaeus 1766) and armadillos (*Dasypus novemcinctus*, Linnaeus 1758) occurred upon 95% of Green (*Chelonia mydas*), Loggerhead (*Caretta caretta*) and Leatherback turtle (*Dermochelys coriacea*) nests laid in Hobe Sound, Florida in 2002 [23], whereas high Loggerhead turtle nest destruction (78%) resulted from nest infestation by beetle larvae (*Lanelater sallei*, LeConte 1853) in southeast Florida from 2002 to 2003 [24]. Furthermore, 22 species of ants have been identified raiding Green, Loggerhead and Leatherback turtle nests in Palm Beach, Florida [25].

According to Cornelius (1982), the most frequent and conspicuous mammal predators of Olive Ridley turtle nests and hatchlings along the Pacific coast of Costa Rica are the coatimundi (*Nasua nasua*), followed by raccoons (*Procyon* spp.) and coyotes (*Canis latrans*). Avian predators were also observed, with magnificent frigate birds (*Fregata magnificens*), black vultures (*Coragyps atratus*), caracaras (*Caracara cheriway*), and turkey vultures (*Cathartes aura*) creating the greatest impact. Additionally, 29 species of birds, mammals, reptiles, and crustaceans were also listed as either suspected or confirmed scavengers or predators of Olive Ridley eggs and/or hatchlings [26].

Sea turtle conservation programs boomed throughout the Pacific coast of Costa Rica during the 90s at different Olive Ridley nesting beaches, to not only monitor nesting activity and estimate population trends [27], but also to protect nests with the intent of increasing hatchling production, thus preventing further population detriment [17,28]. The Rescue Center for Endangered Marine Species (CREMA by its Spanish acronym) has been running a sea turtle nesting monitoring program for over 24 years along the Costa Rican Pacific [12,21]. Although predation and other nesting events have been recorded over the years, these events have not been properly described nor analyzed until recently. Reavis et al. (2022) described the threats that predation and human interactions represent at two Olive Ridley nesting sites monitored by CREMA, with their findings showing similar tendencies and dynamics when compared to other nesting sites, where typically predation and poaching decrease considerably with time after conservation projects are initiated [21].

Corozalito beach is a solitary and nascent arribada nesting beach in the Pacific of Costa Rica, which has been monitored uninterrupted from June to January since 2008 by CREMA. This location holds the highest number of solitary nesting events (over 2000 records per season) among the sites covered by CREMA’s monitoring program [29], and more recently has had estimated numbers in arribadas ranging between 2000 and 21,000 egg-laying females in a single nesting event [12,29]. These mass nesting events normally occur once per month, from August to December, overlapping with solitary nesting events [12]. The aim of this study is to evaluate natural nest predation upon solitary Olive Ridley nesting events (quantity, percentage, distribution of predated and partially predated nests and most common predators). This evaluation intends to provide insights on the impact of predation upon Olive Ridley sea turtle nests at this important site and recommend effective conservation strategies.

## 2. Materials and Methods

### 2.1. Study Site

Corozalito (9°50′54.28″ N, 85°22′43.86″ W), located in the Southern Nicoya Peninsula, Guanacaste, Costa Rica (Figure 1), is a beach flanked to the north and south by rocky outcrops, with an associated wetland and river mouth to the south [12]. For monitoring purposes, only the available nesting area of the beach was divided transversally into 25 m sectors from north to south using numbered posts, which corresponded to 768 m of beach. Sector markers were placed in line with the vegetation, with numbers facing the water [27]. These sectors were then grouped into three main areas: the northern area (sectors 1 to 10), characterized by dense shrubs and low-lying trees, the central area (sectors 11 to 20), characterized primarily by high standing palm trees and little to no ground cover (this area also contains multiple picnic tables and a shelter), and lastly the southern area (sectors 21 to 30), that is primarily ground cover with some mangrove trees encircling the small estuary [12,27]. This beach in the Pacific of Costa Rica shows two seasons: a dry season, or “summer”, from January through June, and a rainy season, or “winter”, from July through December [12].

In addition to the sectors, the beach was divided into three zones that lie parallel to the high tide line, determined by the proximity to either the vegetation and/or the average high tide line (Zone 1 = below the average high tide line; Zone 2 = between average high tide line and vegetation; Zone 3 = above the line of vegetation) [29,30] (Figure 2).

Overall, the beach shows the primordial characteristics of an Olive Ridley sea turtle nesting beach; it is a sandy beach, with a gentle inclination platform, easy access from the sea and an important river/estuary at one of its ends [17,31,32]. Organic debris is abundant on this beach and includes logs and tree branches from the nearby river mouths. Palm trees are predominant, yet beach almond trees (*Terminalia* sp.), button wood mangrove (*Conocarpus erectus*), white mangrove (*Laguncularia racemosa*) and other small shrubs are also common [27]. In addition, Corozalito not only hosts solitary nesting events for Olive Ridley, but also Green, Hawksbill and Leatherback turtles, and the mass nesting events (arribadas) previously mentioned [12,29]. Nesting activity has showed an increase over the 13-year study, with an average of 2156 nesting events per year, with over 45% of nesting occurring on the northern sectors and approximately 60% in zone 2 (CREMA unpublished). Neither predation over arribada nests nor the impact of arribadas were considered on our analysis.

### 2.2. Data Collection and Analysis

Data on sea turtle nest predation upon solitary nesters were compiled following the CREMA monitoring protocol [29]. Field researchers checked the integrity of all nests laid during night patrols or morning surveys from 2008 to 2021 nesting seasons (running from June to January), recording date, time, sector, zone, status of nests (predated or partially predated) and predator (when possible) [21]. A predation event was considered when broken eggshells were found around the nest, with evidence of non-human disruptions [21,33]. Data were taken only on recently predated nests between 19:00 and 05:00 (times when night patrols and morning censuses were conducted) [30,32]. The type of predator was determined by how the nest was entered (digging, poking holes, etc.) and tracks around the nests [21]. The primary predator was determined by the fresher tracks in nests during night patrols, whereas secondary and/or third predators were confirmed during morning surveys on nests that have been previously recorded to be predated at least once [33]. Sometimes, predators were observed searching around freshly laid nests and later on digging the nests to predate on them; when possible, photos were taken [29,33]. Notes on predation were also taken when performing nests excavations [21,29]. Movements or dynamics of predators were not analyzed. In contrast to other CREMA monitoring projects [21], all nests laid in Corozalito were left in situ due to the high number of sea turtles that come to nest, either in solitary fashion or in arribadas [29,30,34]. Data from the 2020 season were collected only during the morning censuses due to the restrictions during the global COVID-19 health emergency.

Descriptive and summary analyses were calculated, including means, standard deviations, minimum and maximum values, using Excel ^®^ 2007 spreadsheets and STATISTICA ^®^ v7 [35]. Predation rates were calculated as the percentage of total nests that were destroyed and/or consumed by predators among the total of nesting events during the study [36]. Due to the non-normal distribution of predated nests as shown by the Shapiro–Wilk test, non-parametric statistical analyses were performed using the Friedman test to detect any associations with their spatial distribution among sectors; statistical significance was set at α ≤ 0.05. This analysis was carried out in program R [37].

## 3. Results

### 3.1. Partial and Complete Predation of Nests

We found a total of 4450 predated nests (partial and complete) from 2008 to 2021 nesting seasons (N = 30,148 nesting events recorded) (Table 1). Only 7.23% (n = 332) were considered partially predated, although predation was most certainly complete after several days. The follow up of partially predated nests, however, was not monitored.

Predation rates showed a fluctuating trend during the sampling period, with increasing values over the years and four distinctive downfalls in 2010, 2014, 2016 and 2017 (Table 1). The lowest predation rate (4.711%) occurred in 2017 and the highest rate during the 2020 nesting season (28.35%). Alternatively, predation rates were the highest in August and September (3.55% and 3.36%, respectively), followed by a decreasing trend by the end of the season (Figure 3).

### 3.2. Distribution of Predation Events

The distribution of predation events alongside the beach showed an heterogenous pattern, with 47.62% of the total predated nests in the northern area of the beach (sectors 1 to 10), followed by the central area (sectors 11 to 20) accounting for 28.67% of predated nests, and lastly the southern area (sectors 21 to 30), with 23.71% of predated nests (Friedman test, chi-squared = 14.778, df = 2, *p*-value = 0.000). Individually, sectors 4 and 5 showed the highest number of predated nests (Sector 4 = 27.86 ± 23.08 predated nests; Sector 5 = 23.36 ± 17.35 predated nests), while sectors 1 and 30 showed the lowest (Sector 1 = 1.93 ± 2.40 predated nests; Sector 30 = 1.00 ± 1.80 predated nests) (Figure 4). Indeed, the pairwise Wilcoxon signed rank test between these three areas revealed statistically significant differences in the number of predated nests between the northern and southern sectors (*p*-value = 0.006), a suggested difference between the center and northern sectors (*p*-value = 0.08) and no significant differences between center and southern sectors (*p*-value = 1.00). In addition, most predated nests were recorded in zone 2 (218.43 ± 170.69; 68.72%), while zone 1 showed the lowest number of predated nests (9.14 ± 13.67; 2.88%).

### 3.3. Nest Predators

Nine species of mammals, birds, crustaceans and insects were identified and confirmed as nest predators (only eggs) in Corozalito (Table 2). In most of the cases (75.92%), nest predators remained unidentified as tracks were erased or were difficult to categorize due to rain, tides, debris, other turtle tracks, tourism, etc., which causes a bias in identifying predators [22,31,38]. Predators were identified in 24.08% of the predated nests (n = 896 predation events). The most active and visible predators were raccoons (55.69%), followed by vultures (22.77%).

One, two or even more predators at a time were recorded predating nests. In most cases (63.90%), two different species were recorded predating a nest as scavengers or secondary predators (mostly hermit crabs and vultures). These predators were observed during morning surveys predating over nests that had been partially predated during night patrols.

## 4. Discussion

### 4.1. Predation Rates

In the past decade, nests from Corozalito have become an easy target for different predators. A noticeable increase in predation rates was detected, from 2008 with 7.95% to 2021 with close to 30% of nests predated. Predation rates in Corozalito showed four dips during the sampling period, with the lowest predation rate occurring in 2017 (Table 1), which is closely related to the total number of solitary nesting events (n = 3396) during that nesting season (CREMA, unpublished). The highest predation rate (28.35%) was recorded in 2020, when beaches and other public places were closed to researchers and the public due to the restrictions imposed by the Costa Rican government during the COVID-19 health emergency. Reavis et al. (2022) reported the predation dynamics of two solitary nesting sites, San Miguel and Costa de Oro, about 15 km south of Corozalito, finding an overall decreasing predation rate trend. They found that the differences regarding predation rates between both sites were primarily due to the length dissimilarity in conservation efforts on each site; with higher predation rates (25%) in the most recent project site (Costa de Oro, starting date 2012), in comparison to San Miguel (6%), with a longer trajectory on protecting sea turtle nests (starting date 1998) [21,27]. Different authors have mentioned that, in the absence of monitoring efforts in nesting beaches, both the number of predators and frequency of predation increases significantly, resulting in changes for both predator and prey alike [23,26,38], a trend that has been observed in Corozalito during the 2020 COVID-19 lockdowns, and in San Miguel and Costa de Oro before the establishment of CREMA’s conservation program.

High nest predation has been observed in other nesting beaches worldwide with high nest densities [20,39,40,41,42,43], where nest predation rates ranging from 30% to 80% have been detected, before and/or during the incubation period [20,38,39,40,43]. Corozalito is a high nest density beach, with up to 2800 solitary nests deposited per year (CREMA, unpublished), as well as several arribadas per year involving several thousands of nesting females over a period of 3 to 4 days [12]. While arribada nesting may reduce the chance for predation due to predator satiation, it has been previously noted that it could also increase overall predation by attracting predators [6]. In fact, the increased predation over time that we found in the present study agrees with the pattern exhibited by Corozalito’s arribada, which displays a clear growing trend in annual frequency and days of duration (from 2008 to 2021) [12]. Predation events were also highest in August and September, followed by a steady decrease in these percentages during the remaining months, which corresponded to the months with the lowest nesting activity in the area. Rojas-Cañizales et al. (2022) suggested that Corozalito is in its early stages of development as an arribada beach, with arribadas occurring from August to December and the largest recording during the months of September and October [12]. These events also coincide with the occurrence of the largest arribada events in the Eastern Tropical Pacific region. These conditions may suggest the hypothesis that arribada beaches attract more predators, which then may have an impact on nesters that choose the arribada beach to nest individually. Several authors have also suggested that predation might play a relatively small role in determining hatching success in arribada beaches, but it may be heavily detrimental in solitary nesting [26,42,44]. In fact, in beaches where both reproductive strategies take place, other authors have reported that 51% of first-night solitary nests were predated in contrast to a low percentage of arribada nests (7.8%), supporting the hypothesis of predator satiation in the evolution of the arribada phenomena [42,44]. Unfortunately, predation rates on arribada nests were not accounted for in this study, hence making it difficult to estimate the overall predation threats upon Olive Ridleys nests (both solitary and arribada) at Corozalito.

### 4.2. Spatial Distribution of Predation Events

Several variables may determine the distribution of predation events alongside the beach. Kolbe and Janzen (2002) explained that there is a significant effect of distance from the edges (water or wooded edges) where the nests are located. In their study, the water edge represented an abrupt ecological edge, which demarcates where resources transition from completely unavailable to available to some degree [45]. Similarly, the environmental characteristics in Corozalito suggests playing an important role in the dynamics of predation. For instance, a larger number of predated nests and, hence, a higher predation rate occurred at the northern sectors (Figure 4), where most nesting events (successful and unsuccessful) have been recorded [30]; this area shows ideal physical characteristics (such as grain size, slope and humidity, among others) for nesting, as described by several authors before [26,46,47]. In addition, the presence of dense vegetation at the northern end of Corozalito might provide an ideal predator habitat and shelter [38,40]. Subsequently, in the central area, the regular visits by community members, tourists, and researchers could have a barrier effect towards predators, excluding them from the nests laid in this area [21]. Lastly, in the southern sectors are covered by rocks, which might represent a major factor that drives nest selection in Olive Ridleys, therefore resulting in lower predation rates [45,48].

Additionally, beach profile results demonstrated that the highest incidence of predation occurred in zone 2 (between average high tide line and vegetation), which may be due to the significantly higher frequency of nesting events that take place in this zone as well [29,40,49,50]. Several authors found similar results, with nests often located on the un-shaded center of the beach with a strong preference for nests to be located as far from the high tide line as possible [47,50,51,52,53]. Even though most nests were placed far from the high tide line, some of them were flooded, damaged, or eroded by sea water (due to ‘spring tides’) and later found to be destroyed by predators.

### 4.3. Predator Identification

According to several authors, some of the most common sea turtle nest predators are feral domestic animals and raccoons [26,38,40]. Cortes-Briceno (2015) recorded nest predation in Mexico, where the complete predation of Green and Hawksbill turtle nests was often associated with feral animals (mainly dogs), while partial predation of nests was caused by other animals such as birds, ghost crabs, and ants (the latter being the most frequent) [38]. Raccoons are the main raiders of nests in the United States, with up to 80% of total Loggerhead and Green turtle nests predated on certain southern beaches [22,54,55]. Reavis et al., 2022, found that the main predators of Olive Ridley nests in San Miguel and Costa de Oro were domestic dogs (*Canis familiaris*), followed by raccoons, which accounted for 58% and 42% of depredated nests, respectively [21].

The most common predators recorded in Corozalito were raccoons (55.69%), which were observed digging up undisturbed nests, leaving them vulnerable to secondary predators and/or scavengers such as vultures (22.77%) and hermit crabs (13.28%). Other predators, such as coatimundis (6.14%) and feral dogs (0.89%), were only recorded predating on well-developed clutches. The main predators and their impacts on the nests are different between Corozalito and the other two project sites (San Miguel and Costa de Oro), which could be due to the proximity of the population centers in these three localities; the town of Corozalito is 2.5 Km inland [56], whereas San Miguel and Costa de Oro communities are just a few steps from the beach [27], making it an easy access for domestic animals. Similar results according to predation richness were found in Playa Nancite (approximately 120 Km north of Corozalito) [31,57]. Nelson and Mo (1996) indicated that nest predation was performed, in decreasing order, by coyotes (*Canis latrans*), raccoons (*Proycon lotor*), coatis (*Nasua narica*), humans (*Homo sapiens sapiens*), and caracaras (*Polyborus plancus*). Both Corozalito and Nancite’s population centers are far inland from the nesting site, with the latter also located in the Santa Rosa National Park, known as a protected area [27,31,33].

Other predators, such as caracaras (0.36%), skunks (0.45%), and tayras (0.12%), were considered rare encounters, perhaps opportunistic predators [20,21,22,40,57]. Finally, maggots were only observed during excavations of old or non-developed nests [22,38].

### 4.4. Conservation Efforts

Nest predation has recently become an issue in Corozalito, which has increased over the years and needs to be addressed with an appropriate approach [58]. In fact, Corozalito is a public beach that was initially protected by a group of inhabitants from the nearby town, many of which have expressed that the increase in nesting sea turtles, and thus nests laid per season, was responsible for the increase of predators that find an endless source of food on this beach [29,58]. Predator impacts on eggs and hatchlings of sea turtles are somewhat easy to document, a necessary task to plan an effective management program [26,59].

Many conservation programs have addressed both nest predation and poaching (illegal taking) by implementing hatcheries, where nests are relocated for their protection, and monitoring programs to decrease and/or regulate the presence of people on the beach. These actions have helped in the conservation of several sea turtle populations worldwide [38,40,60]. However, hatcheries have several limitations and may not always be the best approach [60,61,62]. In high density nesting beaches such as Corozalito, preliminary assessments have concluded that less manipulative options are more practical and effective [20,39,40,41,43], and with the increase in frequency and number of arribadas the beach itself serves as an in-situ hatchery [34]. The importance of Corozalito’s sea turtle nesting populations and their nests are twofold; firstly, as mentioned before, this beach holds up approximately 2800 solitary nesting events of Olive Ridleys each nesting season, which could represent up to 80% of hatching success, that potentially would maintain part of the Eastern Pacific Olive Ridley turtle population [47,61,63]. Secondly, Corozalito is now known as a nascent arribada beach, with higher hatching success of nests laid during arribadas, reaching up 50% [12], three times higher than already established arribada beaches such as Ostional and Nancite [11,12,51]. This could also indicate that the hatching success is higher in solitary nesters regardless of high predation rates, although further research in this aspect is needed.

As described by several authors in different nesting beaches, patrol-based monitoring programs have resulted in a successful reduction of several threats to both nesters and nests [21,23,26,28,31,64].We propose to continue CREMA’s sea turtle nesting monitoring program, along with community-based programs (e.g., environmental education classes, incorporating locals in the monitoring program, among others), which will ultimately involve residents and visitors in the conservation of these reptiles [21,26,28,64]. It is important to include predation rates from other sea turtle species’ nests, as well as predation both during solitary and mass nesting events. Similarly, other threats, such as illegal egg extraction (poaching), have not been addressed since CREMA’s nesting monitoring program started in 2008 [33]. Viejobueno et al. (2012) estimated egg extraction rates in Corozalito from 2008 to 2011, observing a decreasing trend of poaching events from 30% to 6% during the three-year study [33]. Nine years later, a study on illegal egg extraction in several localities both in the Caribbean and the Pacific coasts of Costa Rica tracked a poached turtle nest from Corozalito to a supermarket in the Central Valley of Costa Rica, 137 km away [65]. Today, predation rates swivel around with similar values (5% in 2022, CREMA’s unpublished data), nonetheless the general impact over nesters and nests has not been evaluated. Additionally, climate change must be assessed in order to give a complete report of all the threats that these species are facing in Corozalito and the impact on their populations [12,33]. Long-term analysis of the nesting population on this beach would be beneficial in the future to elucidate the effects of all threats and conservation efforts [23]. The design of a proper beach management plan is an imperative and valuable conservation strategy for the nesting sea turtles of Corozalito [12], which must include potential alternatives to manage predators’ impact (e.g., translocation, altering predator foraging strategies [66], among others). These measures could help to reduce both human and natural loss of nests and improve measures to safeguard the Olive Ridleys and their nesting habitat [59,67]. For lasting results, such alternatives must be supported and promoted by several entities such as the Environment and Energy Ministry, local communities, NGOs, other state institutions, and academia, among others.

## 5. Conclusions

Our results suggest a clear increase in the predation of Olive Ridley turtle nests throughout the 13-year study period. This trend, although fluctuating over the years, could be also an indication of the increasing number of nesting events in Corozalito, and its full protection might be facilitated by the designation of Corozalito as a National Wildlife Refuge with a corresponding marine protected area, which together with the drive of the surrounding coastal communities [59] and a continuation of long-term monitoring will better elucidate the possible effects on the nesting abundance of Olive Ridleys.

## Figures and Tables

**Figure 1 animals-13-00875-f001:**
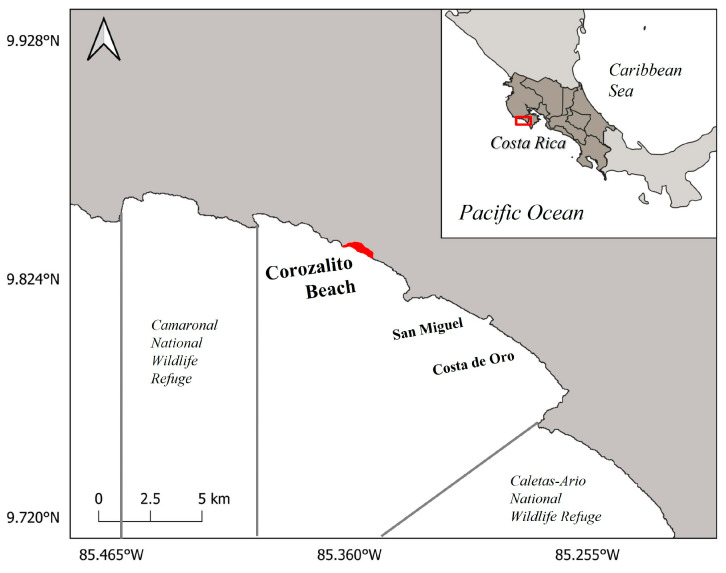
Geographical location of Corozalito and the surrounding Marine Protected Areas of the Camaronal and the Caletas-Arío National Wildlife Refuges, southern Nicoya Peninsula, Costa Rica. Bold red represents the monitored area.

**Figure 2 animals-13-00875-f002:**
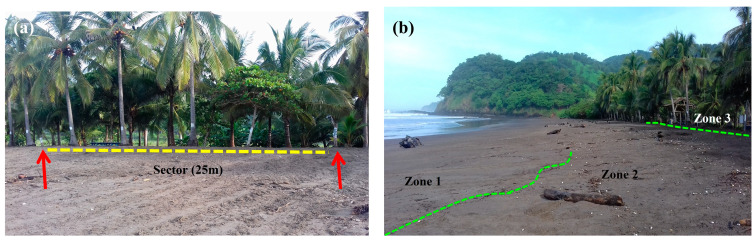
Division of the beach according to (**a**) sectors (markers indicated by arrows), and (**b**) zones in Corozalito (Photos: Espinoza-Rodriguez, 2019).

**Figure 3 animals-13-00875-f003:**
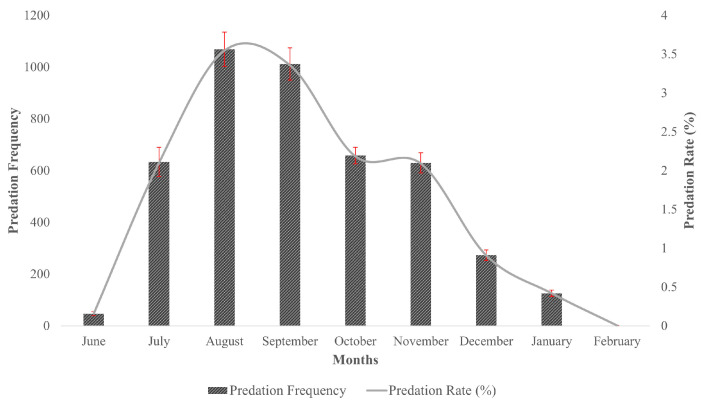
Monthly predation rates (%) in Corozalito during the sampling period (2008–2021).

**Figure 4 animals-13-00875-f004:**
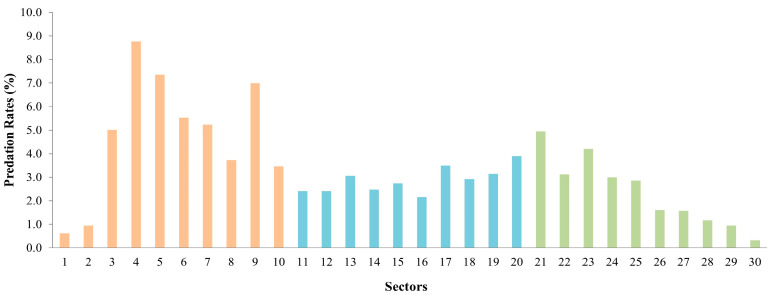
Percentage of predated nests per sector in Corozalito, from 2008 to 2021; Northern Area (orange) = sectors 1–10, Central Area (blue) = sectors 11–20, Southern Area (green) = sectors 21–30.

**Table 1 animals-13-00875-t001:** Predation Rates (%) of Olive Ridley sea turtle nests in Corozalito, from 2008 to 2021. Percentage values are relative to the total number of nesting events. Total predation events include both partial and complete predated nests.

Season	Total Nesting Events	Total Predation Events	Annual Predation Rate (%)
2008	1119	89	7.95
2009	1782	162	9.09
2010	1763	98	5.56
2011	1512	150	9.92
2012	1588	275	17.32
2013	1772	453	25.56
2014	1918	107	5.58
2015	2923	445	15.22
2016	2403	157	6.53
2017	3396	160	4.71
2018	2313	489	21.14
2019	2656	500	18.83
2020	2303	653	28.35
2021	2700	712	26.37
Total	30,148	4450	14.80

**Table 2 animals-13-00875-t002:** Olive Ridley sea turtle egg predators and scavengers identified at Corozalito during the sampling period.

Class	Scientific Name	Common Name (% of Occurrence)
Mammals	*Procyon* spp. *Nasua narica* *Conepatus semitriatus* *Eira barbara* *Didelphis marsupialis* *Canis lupus familiaris*	Raccoon (55.69%) Coati (6.14%) Stripped Skunk (0.45%) Tayra (0.12%) Common Opossum (0.05%) Dog (0.89%)
Birds	*Caragyps atratus* *Caracara cheriway*	Black Vulture (22.77%) Caracara (0.36%)
Crustaceans	*Coenobitidae compressus*	Ecuadorian Hermit Crab (13.28%)
Insects	Unidentified	Maggots (0.12%)

## Data Availability

Restrictions apply to the availability of these data. Data were obtained from CREMA’s dataset and are available from the authors with the permission of Isabel Naranjo (Director of CREMA).

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
