# Peer review of "Predation Rate on Olive Riley Sea Turtle (Lepidochelys olivacea) Nests with Solitary Nesting Activity from 2008 to 2021 at Corozalito, Costa Rica"

_animals, 2023, doi:10.3390/ani13050875_

Round 1

Reviewer 1 Report

The MS presented titled "Predation rate on Olive Ridley sea turtle (Lepidochelys olivacea) nests with solitary nesting activity from 2008 to 2021 at Corozalito, Costa Rica" has great relevance in the knowledge of these turtles and in particular what happens with solitary nesting on little-known beaches and that also requires a study of great importance because it can be compared and know-how are the measurements of security for the protection on said beaches. Which can be replicated on other beaches or with other species of turtles in that country or in others.

Line 11. Was the entire 768 m long beach marked like this? explain

Line 118. To understand what will be discussed later, describe the times of the year considered to be rainy and dry.

Line 139. How was the estimate made? Explain.

Line 152. Only by species of predators and not by the number of each species? Explain

Line 212. Is there a percentage of mixed predation per nest?

Line. 219. Could it be 4, not 3?  2010, 2014, 2016, and 2017? What were the bases to define only 3 and no more? Explain.

Line 250. How the rainy season or natural disasters such as hurricanes and earthquakes impact

Line 265. How does nesting influence the arrival and loss of nests by other turtles nesting at the same time?

Line 308. No problems with the feral dog population? 

Line 343. Explain the causes and how they are related on that beach.

Line 345. On that beach, what is the idea of some nests would be protected?

Line 351. Although it is understood that massive arrivals provide enough food, explain how lonely arrivals affect or benefit other areas.

Line 354. Is it known what the density of predators is throughout the year and what happens at times of the year when there are no high rates of predators, do they move to other beaches or only their population decreases? Explain

Line 359. This is important, but you could explain how you recruit more volunteers or local people to achieve this goal.

Line 366. What is the impact of poaching in this area, are there reports of that? Explain.
